# Is CCL2 an Important Mediator of Mast Cell–Tumor Cell Interactions in Oral Squamous Cell Carcinoma?

**DOI:** 10.3390/ijms24043641

**Published:** 2023-02-11

**Authors:** Bernhard Hemmerlein, Luisa Reinhardt, Bernhard Wiechens, Tatjana Khromov, Henning Schliephake, Phillipp Brockmeyer

**Affiliations:** 1Institute of Pathology, Helios Klinikum Krefeld, 47805 Krefeld, Germany; 2Department of Oral and Maxillofacial Surgery, University Medical Center Goettingen, 37075 Goettingen, Germany; 3Department of Orthodontics, University Medical Center Goettingen, 37075 Goettingen, Germany; 4Institute for Clinical Chemistry, University Medical Center Goettingen, 37075 Goettingen, Germany

**Keywords:** mast cells, MCs, oral squamous cell carcinoma, OSCC, soluble factors, CC chemokine ligand 2, CCL2, MCP-1

## Abstract

In this study, we aimed to evaluate the influence of interactions between mast cells (MCs) and oral squamous cell carcinoma (OSCC) tumor cells on tumor proliferation and invasion rates and identify soluble factors mediating this crosstalk. To this end, MC/OSCC interactions were characterized using the human MC cell line LUVA and the human OSCC cell line PCI-13. The influence of an MC-conditioned (MCM) medium and MC/OSCC co-cultures on the proliferative and invasive properties of the tumor cells was investigated, and the most interesting soluble factors were identified by multiplex ELISA analysis. LUVA/PCI-13 co-cultures increased tumor cell proliferation significantly (*p* = 0.0164). MCM reduced PCI-13 cell invasion significantly (*p* = 0.0010). CC chemokine ligand 2 (CCL2) secretion could be detected in PCI-13 monocultures and be significantly (*p* = 0.0161) increased by LUVA/PCI-13 co-cultures. In summary, the MC/OSCC interaction influences tumor cell characteristics, and CCL2 could be identified as a possible mediator.

## 1. Introduction

Mast cells (MCs) are potent effector cells of the immune system involved in numerous physiological and pathological conditions, such as angiogenesis, tissue remodeling, wound healing, IgE-dependent allergic disease, infection-induced immune responses, and autoimmune inflammatory disease [1].

Depending on the tumor entity and MC localization (intratumoral vs. tumor stroma), tumor-promoting and tumor-inhibiting effects have been described. Evidence also suggests that MCs are involved in the progression of various malignomas. Thus, MCs in the stroma of malignant melanoma, pancreatic carcinoma, or gastrointestinal adenocarcinoma are associated with poor patient prognosis [2]. In contrast, a tumor-suppressive effect has been demonstrated for intratumoral MCs in prostate carcinoma [3].

During tumor progression, MCs accumulate near blood vessels located around the tumor, secreting several mediators promoting angiogenesis and suppressing the immune response, including vascular endothelial growth factor (VEGF), histamine, tumor necrosis factors (TNF-𝛼), and interleukin 18 (IL-18) [4]. Similarly, MC-specific proteases such as chymase and tryptase (MCT) are crucial for degrading the extracellular matrix (ECM) and inducing angiogenesis, thereby promoting tumor progression [4]. Stem cell factor (SCF) is critical in the MC/tumor cell interaction. Therefore, SCF expressed by tumor cells promotes MC migration and activation, leading to an increased release of more SCF molecules by tumor cells in a positive feedback loop [5].

In addition to their importance in local tumor progression, MCs are involved in lymph node metastasis (LNM) [6]. Hence, it has been demonstrated that the number of MCT-positive (MCT^+^) intratumoral MCs positively correlates with the number of emerging LNMs in gastrointestinal malignancies. These properties additionally attribute a special role to MCs as biomarkers for prognosis assessment and as potential targets for future tumor therapies [4,6].

In addition to their effect on tumor progression and metastasis, MCs seem relevant in drug-based tumor therapy. Immune checkpoint inhibitors (ICIs) may experience reduced efficacy due to the presence of MCs. In a mouse model, a high intratumoral MC density was associated with a lower efficacy of immunotherapies targeting programmed cell death protein 1 (PD-1) inhibitors in melanoma cells [7]. Combining tyrosine kinase inhibitors, such as sunitinib or imatinib, which target MCs, substantially increases the response to therapy with PD-1 inhibitors in mice [7]. High MC density has also been associated with increased resistance to therapy in other malignancies, such as breast or prostate tumors [3,8].

In the oral cavity, a high MC density mainly aids inflammatory processes in diseases such as gingivitis or periodontitis [9]. The relevance of tumor-associated MCs in oral squamous cell carcinoma (OSCC) progression has been controversially described [9]. Angiogenesis plays a crucial role in OSCC progression and promotes local spread and metastasis [10]. It has been shown that more blood vessels and MCs are found in the environment of OSCC compared to healthy oral mucosa, supporting the hypothesis that MCs have a tumor-promoting effect in OSCC by inducing angiogenesis [10]. In contrast, MC density has recently been shown to decrease from oral potentially malignant disorders (OPMD) to OSCC [11], and increased vascularity in OSCC appears to be inversely related to MC density [12,13]. These findings suggest a protective role of MCs in OSCC. Increased MC density has also been described in oral leukoplakia, which may be associated with the progression toward invasive carcinoma [14]. Various studies have shown a tumor-promoting effect through the induction of angiogenesis and degradation of the ECM. MC density has been associated with increased tumor progression in lip carcinoma [15] and with a poorer prognosis in tongue malignancies [16]. However, a low MC density in the stroma of OSCC was associated with poorer prognoses for patients compared to those with a high MC density [17]. In line with these results, we recently showed in many patient tissue samples that a high MC density in the tumor-associated stroma of OSCC is positively correlated with longer overall survival (OS) and reduced MC degranulation [18].

Although the influence of MCs on OSCC progression is already known, it is still unclear which molecules mediate MC/OSCC interactions. In this study, we investigated MC/OSCC interactions, considering the influence on tumor cell proliferation and invasion and identified CC chemokine ligand 2 (CCL2) as a potential interaction mediator.

## 2. Results

### 2.1. Influence of the MC/OSCC Interaction on Tumor Cell Proliferation

To investigate the influence of the MC/OSCC interaction on tumor cell proliferative behavior, OSCC cell line PCI-13 was cultured in LUVA (MC)-conditioned medium (MCM), and LUVA/PCI-13 co-cultures were performed. Proliferation was measured using a BrdU uptake assay. Untreated PCI-13 cells served as controls. Data show that PCI-13 cell proliferation increased when treated with MCM compared with untreated controls. A significant (*p* = 0.0164) increase in PCI-13 cell proliferation was demonstrated for LUVA/PCI-13 co-cultivation compared to the untreated controls (Figure 1A).

In summary, a significant effect of direct MC/OSCC interactions on PCI-13 cell proliferation was observed (*p* = 0.0164).

### 2.2. Influence of the MC/OSCC Interaction on Tumor Cell Invasion

PCI-13 cells were cultured in MCM, and tumor cell invasion assays were performed to investigate the effects of the MC/OSCC interaction on tumor cell invasive behavior. PCI-13 cells cultured in an untreated medium served as controls. The invasion analysis data revealed a significant (*p* = 0.0010) reduction in PCI-13 tumor cell invasion compared to untreated controls (Figure 1B).

In conclusion, a significant effect of the indirect OSCC cell stimulation on PCI-13 tumor cell invasion was observed (*p* = 0.0010).

### 2.3. Identification of Known MC/Tumor Cell Mediators

An individual multiplex ELISA assay was performed with MC/tumor cell interaction partners already known from the literature [19] to identify potent factors in the conditioned media mediating the MC/OSCC interaction: chemokine (C-X-C motif) ligand 1 (CXCL1), interleukin 6 (IL-6), interleukin 8 (IL-8), interleukin 10 (IL-10), chemokine (C-C motif) ligand 3 (CCL3), matrix metalloproteinase 2 (MMP-2), matrix metalloproteinase 9 (MMP-9), OX40 ligand (OX40L), CC chemokine ligand 5 (CCL5), stem cell factor (SCF), transforming growth factor beta 1 (TGF-β1), tumor necrosis factor alpha (TNF-𝛼), vascular endothelial growth factor (VEGF), and CC chemokine ligand 2 (CCL2). Conditioned media of PCI-13 and LUVA monocultures and LUVA/PCI-13 co-cultures were compared accordingly. Unconditioned cell culture media was the control.

Of the soluble factors mentioned above, significant release differences could only be detected for CCL2, TNF-𝛼, and VEGF, shown in Figure 1C (all *p*-values < 0.05). CCL2 was not detectable in the control media and was only detected to a minor extent in the PCI-13- and LUVA-conditioned media. However, a significant increase in CCL2 release was detected in the co-cultures, compared with the PCI-13- and LUVA-conditioned media (*p* = 0.0161 and *p* = 0.0035, respectively; Figure 1C). While TNF-𝛼 was undetectable in the control media, the high release could be detected in PCI-13-conditioned media and a low release in LUVA-conditioned media. However, co-culturing did not significantly increase or decrease TNF-𝛼 release in either cell line (Figure 1C). VEGF could not be detected in the control media, but a strong release was detectable in the tumor cell-conditioned media and a low release in the MC-conditioned media. However, no significant increase or decrease in VEGF release could be detected in the co-cultures compared with the tumor cell-only-conditioned media (Figure 1C).

In conclusion, a significant increase in CCL2 release could be detected by MC/OSCC co-culturing (all *p*-values < 0.05), suggesting that this factor may be a mediator in the MC/OSCC interaction.

## 3. Discussion

Despite advances in diagnostic and therapeutic options, the prognosis of patients with high OSCC stages has hardly improved over the past decades [20]. MC density in the tumor microenvironment (TME) of OSCC influences the OS and thus could be an interesting approach for targeted tumor therapy [18]. However, the factors mediating MC/OSCC interactions are far from known. In this study, we investigated the effect of direct and indirect MC stimulation on the proliferative and invasive behavior of OSCC tumor cells in vitro. We identified CCL2 as a potential mediator in OSCC for the first time.

The TME, with its numerous tumor-infiltrating host cells, is critical for disease progression and metastasis [21]. Mast cells comprise a large proportion of immune cells in the TME [22]. However, in several studies, precise MC localization has already been defined as a critical factor for the effect of MCs on tumor cells and patient survival. Therefore, high MC density in TME leads to a more prognostically favorable outcome because MCs inhibit tumor expansion and invasion [2]. In OSCC, different effects have been described for the same MC localization patterns, ranging from antitumor to tumor-promoting [15,17,23]. Iamaroon and colleagues analyzed the association between MCs and angiogenesis in OSCC and premalignant dysplasia by immunohistochemistry [23]. The authors demonstrated an association between high MC density and OSCC progression and invasion, which were also positively correlated with increased vascular sprouting in tumor tissue [23]. Rojas et al. described a tumor-promoting MC effect in lip carcinoma [15]. Moreover, healthy and premalignant altered lip tissue was analyzed for MC density by immunohistochemistry. The analysis confirmed increased immune cells and blood vessel density in carcinomas compared to healthy lip tissue [12]. In contrast, Attramadal et al. demonstrated that low MC density was associated with decreased OS in OSCC patients [17]. An immunohistochemical analysis was performed on 62 tissue samples from patients with T stages 1 and 2 OSCC. The MC number in the immediate tumor-surrounding stroma indicated a tumor-suppressive effect of MC in OSCC [17].

MCs have the ability to promote tumor proliferation and invasion both by direct stimulation and indirectly by modulating the TME [24]. The mitogenic activity of tumor cells is partially dependent on the secretion of MC proteases, such as tryptase and chymase [24]. In this study, we confirmed this direct effect through MC/OSCC co-cultivation. However, our studies of indirect MC stimulation of OSCC invasion suggest a reciprocal effect. Thus, different effects appear to occur in the interaction between MCs and OSCC, highlighting the opposing roles of MCs in tumor progression and complicating our current understanding.

It has been described that CCL2 is able to trigger tumor inflammation. TME is a major cause of immunosuppression, while CCL2 is the most potent chemoattractant in macrophage recruitment and a potent trigger of chronic inflammation [21]. Hence, various proinflammatory, stimulatory and signaling molecules, such as TNF-𝛼, can transcriptionally activate CCL2, and oncogenes such as p53 and RB directly regulate CCL2 expression [21]. Moreover, tumor cells can actively secrete CCL2. This CCL2 is responsible for recruiting various immune effector cells, such as macrophages, myeloid-derived suppressor cells (MDSCs), mesenchymal stem cells (MSCs), and regulatory T cells (Tregs) [21]. A mouse model study demonstrated that the CCL2/CCR2 axis is involved in MC recruitment during the inflammatory response [25]. MC attraction via CCL2 has also been demonstrated for pancreatic tumors [26].

In OSCC, we demonstrated for the first time a possible link between MCs and tumor cells via CCL2 as a mediator, consistent with these results. The CCL2 increase by co-cultivation seems parallel to the increase in tumor cell proliferation, suggesting a possible autocrine tumor cell activation. Indirect MC stimulation resulted in a reduction in tumor cell invasiveness. Therefore, as mentioned, MC localization (in direct tumor cell contact vs. more distant in the TME) exerts a differential influence in OSCC. In our opinion, CCL2 may be involved in MC recruitment and could be an interesting starting point for targeted antitumor therapy. However, further studies are needed to understand the exact functional relationships better.

## 4. Materials and Methods

All further described measurements were performed in triplicate in three independent experiments for each cell line.

### 4.1. Cell Culture

Human OSCC cell line PCI-13 was obtained from the University of Pittsburgh Cancer Institute (UPCI, Pittsburgh, PA, USA) [27]. CD34+, c-kit+, and CD13+ human MC cell line LUVA was obtained from KERAFAST (Kerafast, Boston, MA, USA). PCI-13 cells were cultured in MEM with Earle’s salts, 2.2 g/L NaHCO3, stable glutamine, and low endotoxin (Merck, Darmstadt, Germany), supplemented with 10% heat-inactivated fetal bovine serum (FBS) (Biochrom, Berlin, Germany), 1% nonessential amino acids (NEAA) (PAN Biotech, Aidenbach, Germany), 100 U/mL penicillin, and 100 μg/mL streptomycin (PAN Biotech), and incubated in a humidified chamber with 5% CO_2_ at 37 °C. Human MC suspension cell line LUVA was cultivated in 20 mL of StemPro™-34 SFM medium (Thermo Fisher Scientific, Waltham, MA, USA) in upright T175 flasks and incubated in a humidified chamber with 5% CO_2_ at 37 °C.

### 4.2. Preparation of Conditioned Cell Culture Media

An amount of 5 × 10^3^ cells of the PCI-13 cell line was transferred to 10 mL StemPro™-34 SFM medium (Thermo Fisher Scientific, Waltham, MA, USA) and cultured in T175 flasks. After incubating for 48 h in a humidified chamber with 5% CO_2_ at 37 °C, 20 mL of the stem cell medium per bottle were harvested and transferred to a Falcon tube. Cells remaining in the medium were removed by centrifugation at 1250 rpm at room temperature for 5 min. The medium was stored at −80 °C in 20 mL aliquots. An amount of 5 × 10^5^ cells of the MC cell line LUVA was cultured in 20 mL of StemPro™-34 SFM medium (Thermo Fisher Scientific, Waltham, MA, USA). After 24 h, the medium was replaced with a fresh stem cell medium. Cells were cultivated in a humidified chamber with 5% CO_2_ at 37 °C. Conditioned medium was harvested, transferred into a Falcon tube, and centrifuged for 5 min at 1000 rpm at room temperature. The supernatant was separated from the LUVA cell pellet and stored at −80 °C in 20 mL aliquots.

### 4.3. Mast Cell–Tumor Cell Co-Cultures

OSCC cell line PCI-13 was transferred into StemPro™-34 SFM medium (Thermo Fisher Scientific, Waltham, MA, USA) and cultured in T175 flasks to confluence (minimum 48 h incubation, Figure 2A). Twenty-four hours after attachment, the medium was replaced with 20 mL of StemPro™ medium, and 5 × 10^3^ LUVA cells were added per flask. Medium and 5 × 10^3^ cells were harvested and served as the control. The rest of the PCI-13 cells were divided (5 × 10^3^ cells) into new T175 flasks. Cells were co-cultured, ranging from 48 to 72 h (Figure 2B). The medium was collected and centrifuged at 200× *g* for 10 min at room temperature to collect LUVA cells grown in suspension. After collecting LUVA cells by separation from the medium, the PCI-13 cell line growing in monolayer was trypsinized and collected as cell pellets. Collected media (control and after co-culture) were centrifuged at 750× *g* for 10 min, transferred into new falcon tubes, and again centrifuged at 1500× *g* for 10 min. All media and cell pellets were stored at −80 °C in aliquots.

### 4.4. BrdU Cell Proliferation Assay

Amounts of 5 × 10^3^ cells of the PCI-13 cell line and 1.5 × 10^4^ cells of the LUVA cell line were seeded per well of a 96-well plate in 100 µL cell culture medium and incubated for 24 h. Cells were incubated for 4 h with bromodeoxyuridine (BrdU) at a final concentration of 10 µL. According to the manufacturer’s instructions, proliferation analyses were performed using a BrdU-Cell Proliferation ELISA kit (Roche, Penzberg, Germany). The absorbance was measured using a microplate reader (BioRad, Feldkirchen, Germany) at 450 nm (control wavelength 655 nm).

### 4.5. Tumor Cell Invasion Assays

Corning^®®^ BioCoat™ growth factor reduced matrigel invasion chamber with an 8.0 µm PET membrane and Corning^®®^ BioCoat™ control inserts with an 8.0 µm PET membrane were used (Corning, Corning, NY, USA). Before the experiment, 500 µL of warm (37 °C) serum-free culture medium was added to the interior lumen of the inserts and incubated for 2 h at 37 °C for rehydration. The medium was carefully removed from the inserts, and 7 × 10^4^ cells in a culture medium containing 0.1% BSA were added per insert. After 10 min incubation at 37 °C, 600 µL of chemoattractant (20% FBS in culture medium) was added to each well. Then, cells were incubated for 24 h at 37 °C in a humidified chamber with 5% CO_2_. After 24 h, the medium and the remaining cells that had not migrated through the membrane pores were removed using cotton tip swabs. Inserts were incubated with 70% ethanol for 10 min to allow for cell fixation. Ethanol was removed, and the membrane was dried at room temperature for 15 min. Inserts were incubated with DAPI in 0.1% Triton X-100/PBS for 10 min at room temperature to visualize cells that had migrated through the membrane. Inserts were dipped twice in PBS and left in fresh PBS in wells, and three pictures of different areas of the membrane in each insert were taken using a fluorescence microscope (Zeiss, Oberkochen, Germany). Based on pictures of stained nuclei, the number of migrated cells was semi-automatically counted using the FIJI software [28] (Figure 3).

### 4.6. Multiplex ELISA Array

An individual multiplex ELISA assay (Quantibody; RayBiotech Inc., Peachtree Corners, GA, USA) was designed with MC/tumor cell interaction partners already known from the literature [19]: CXCL1, IL-6, IL-8, IL-10, CCL3, MMP-2, MMP-9, OX40L, CCL5, stem cell factor (SCF), TGF-β1, TNF-𝛼, VEGF, and CCL2.

The conditioned media of the PCI-13 and mast cell lines (LUVA) and LUVA/PCI-13 co-cultures were compared accordingly. The concentration of each soluble factor was determined by RayBiotech Inc. using their Quantibody service. An unconditioned cell culture medium served as the control.

### 4.7. Statistical Analysis

Statistical analysis of all in vitro experiments was performed using independent Student’s *t*-tests and one-way ANOVA. All tests were performed at a significance level of α = 5% using the Prism 9 statistical software (GraphPad, La Jolla, CA, USA).

## 5. Conclusions

MC/OSCC interactions affect tumor cell characteristics, and thus, tumor progression, making them interesting candidates for targeted tumor therapy. In this study, we identified CCL2 for the first time in OSCC as a potential mediator of this interaction. Data suggest that CCL2 may promote tumor cell proliferation. Further studies should characterize the functional relationships.

## Figures and Tables

**Figure 1 ijms-24-03641-f001:**
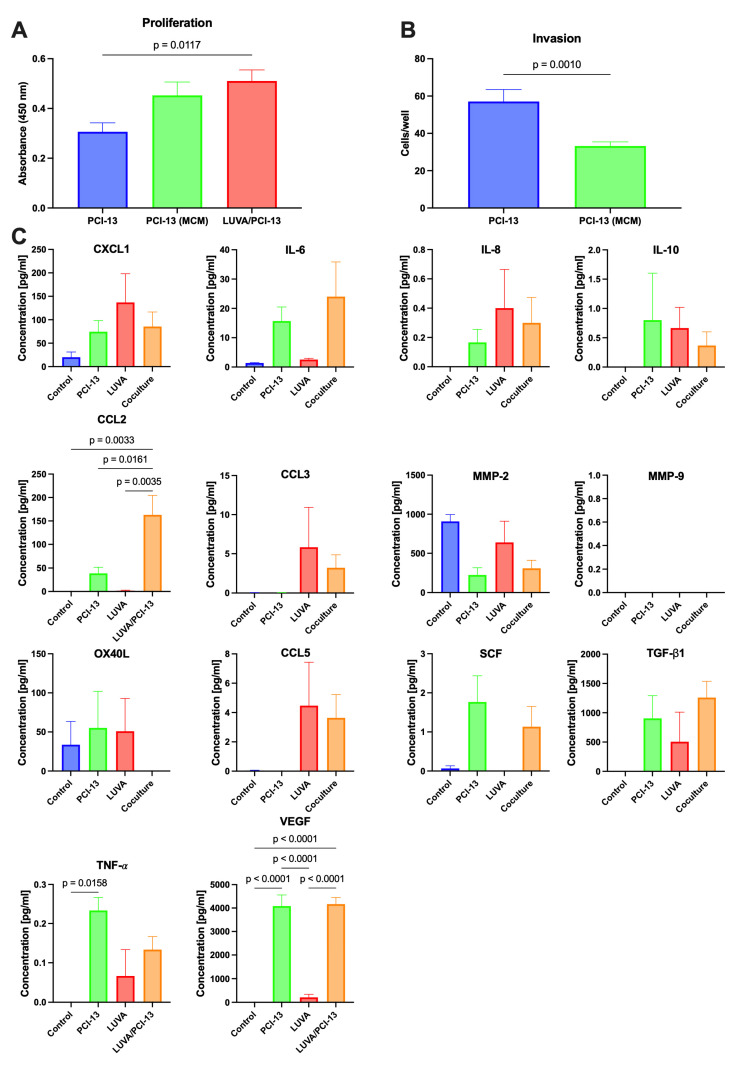
In vitro analysis of MC/OSCC interactions: (**A**) Influence on OSCC cell proliferation. PCI-13 cells cultured in unconditioned medium (PCI-13), PCI-13 cultured in MC-conditioned medium (MCM), and LUVA/PCI-13 co-cultures, based on 5 × 10^3^ OSCC cells, respectively. (**B**) Influence on OSCC cell invasion. PCI-13 cells cultured in unconditioned medium (PCI-13) and PCI-13 cells cultured in MCM. (**C**) Multiplex ELISA analysis of known MC/tumor cell interaction partners: CXCL1, IL-6, IL-8, IL-10, CCL2, CCL3, MMP-2, MMP-9, OX40L, CCL5, SCF, TGF-β1, TNF-𝛼, and VEGF. Cell medium alone as control (control), PCI-13 conditioned medium (PCI-13), LUVA conditioned medium (LUVA), and medium from LUVA/PCI-13 co-cultures (LUVA/OCI-13).

**Figure 2 ijms-24-03641-f002:**
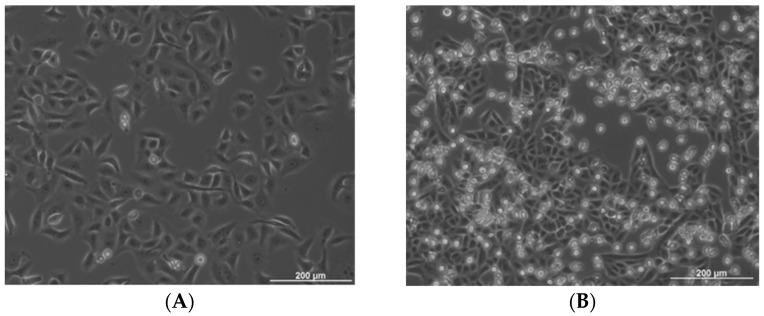
Cell line cultivation: (**A**) PCI-13 monoculture and (**B**) LUVA/PCI-13 co-culture.

**Figure 3 ijms-24-03641-f003:**
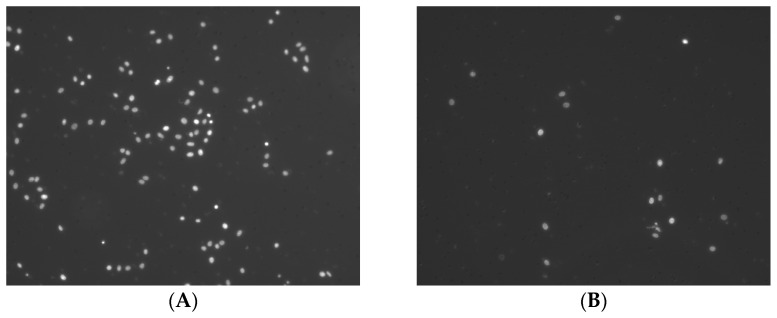
Tumor cell invasion: (**A**) Invasive PCI-13 cells without indirect MC stimulation and (**B**) Invasive PCI-13 cells after indirect MC stimulation (MCM).

## Data Availability

All data are available through the corresponding author.

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
