# Peer review of "Is CCL2 an Important Mediator of Mast Cell–Tumor Cell Interactions in Oral Squamous Cell Carcinoma?"

_ijms, 2023, doi:10.3390/ijms24043641_

Round 1

Reviewer 1 Report

Research Communication “Is CCL2 an important mediator of mast cell-tumor cell interactions in oral squamous cell carcinoma?” by Hemmerlein et. al., identifies CCL2 as a potential mediator in OSCC progression. In this study, the authors investigated the interactions between mast cells (MCs) and oral squamous cell carcinoma (OSCC) cells. They have studied tumor cells and mast cells interaction to assess the rate of tumor proliferation, and invasion rates and further to identify the soluble factors that may help in mediating the crosstalk. The research finding indicate a direct interaction between the human MC cell line LUVA and the human OSCC cell line PCI-13. Results show that co-culturing LUVA/PCI-13 increased the rate of tumor cell proliferation significantly and MCM caused significantly reduced invasion ability of PCI-13 cell. It’s interesting to observe that CCL2 levels increased significantly after MC/OSCC co-culturing and it may be possible that CCL2 could be a possible mediator of this interaction.

The research article can be accepted in its current form. The result, material and methods section provide necessary details related to the culture condition, processing, and statistical analysis.

Author Response

  1. The research article can be accepted in its current form. The result, material and methods section provide necessary details related to the culture condition, processing, and statistical analysis
    1. Authors' response: We would like to thank the reviewer very much for the evaluation and for the kind praise of our work.

Reviewer 2 Report

It is a very interesting study and the results are clear and contribute to a better understanding of the interaction between inflammation and cancer, especially of the oral cavity. However I have some comments and suggestions.

Introduction: Most of the bibliographic citations point to a role of MC in the progression of OSCC and very few to an inhibitory role. I suggest including other references for better understanding of the controversy of the subject.

A bibliographic citation is not included to support the reference to the role of MC in gingivitis and periodontitis (line 62-63). It is relevant because of the relationship between peridontitis and OSCC,

I suggest including the full name of CCL2 and the acronym in parentheses at the beginning of the text.

Discussion: It would be relevant for the authors to explain the different role that MC has in proliferation and invasiveness. How do their results contribute to elucidate this phenomenon? It would be relevant for the authors to explain the different role that MC has in proliferation and invasiveness. How do their results contribute to elucidate this phenomenon?

Conclusions: Perhaps the authors could suggest, according to their results, whether the mediating role of CCL2 is rather activator or inhibitor of OSCC progression.

Author Response

Introduction: Most of the bibliographic citations point to a role of MC in the progression of OSCC and very few to an inhibitory I suggest including other references for better understanding of the controversy of the subject.

Authors’ answer: Thank you for this note to improve the paper. We have added three recent studies to the manuscript that suggest a protective MC-function in OSCC:

Shrestha, A.; Keshwar, S.; Raut, T. Evaluation of Mast Cells in Oral Potentially Malignant Disorders and Oral Squamous Cell Carcinoma. International Journal of Dentistry 2021, 2021.

Ansari, F.; Asif, M.; Kiani, M.N.; Zareef, A.; Rashid, F.; ud Din, H.; Ansar, F. Relationship between Mast Cell Density and Microvessel Density in Oral Squamous Cell Carcinoma and Normal Oral Mucosa: Immunohistochemical Analysis using CD117 and CD34 Antibodies. International Journal of Pathology 2022, 114-120.

Teofilo, C.R.; Ferreira Junior, A.E.C.; Batista, A.C.; Fechini Jamacaru, F.V.; Sousa, F.B.; Lima Mota, M.R.; Silva, M.F.E.; Barros Silva, P.G.; Alves, A. Mast Cells and Blood Vessels Profile in Oral Carcinogenesis: An Immunohistochemistry Study. Asian Pac J Cancer Prev 2020, 21, 1097-1102, doi:10.31557/APJCP.2020.21.4.1097.

A bibliographic citation is not included to support the reference to the role of MC in gingivitis and periodontitis (line 62-63). It is relevant because of the relationship between periodontitis and OSCC

Authors’ answer: Thank you for your comment, we have added a corresponding review article to the manuscript:

Gaje, P.N.; Amalia Ceausu, R.; Jitariu, A.; Stratul, S.I.; Rusu, L.C.; Popovici, R.A.; Raica, M. Mast Cells: Key Players in the Shadow in Oral Inflammation and in Squamous Cell Carcinoma of the Oral Cavity. Biomed Res Int 2016, 2016, 9235080, doi:10.1155/2016/9235080.

I suggest including the full name of CCL2 and the acronym in parentheses at the beginning of the text.

Authors’ answer: Thank you for your comment, we have made appropriate changes in the manuscript.

Discussion: It would be relevant for the authors to explain the different role that MC has in proliferation and invasiveness. How do their results contribute to elucidate this phenomenon?

Authors’ response: Thank you for this useful comment. We have revised the discussion section in the manuscript and included a new reference:

Khazaie, K.; Blatner, N.R.; Khan, M.W.; Gounari, F.; Gounaris, E.; Dennis, K.; Bonertz, A.; Tsai, F.-N.; Strouch, M.J.; Cheon, E. The significant role of mast cells in cancer. Cancer and Metastasis Reviews 2011, 30, 45-60.

Conclusions: Perhaps the authors could suggest, according to their results, whether the mediating role of CCL2 is rather activator or inhibitor of OSCC progression.

Authors’ response: Thank you for pointing that out. Our data suggest that CCL2 may be involved in promoting tumor cell proliferation. We have added appropriate references in the Discussion and Conclusions section. The changes can be found in the manuscript.